# GuttaFlow^®^ Bioseal Cytotoxicity Assessment: In Vitro Study

**DOI:** 10.3390/molecules25184297

**Published:** 2020-09-19

**Authors:** Inês Ferreira, Mafalda Laranjo, Carlos Miguel Marto, João Casalta-Lopes, Beatriz Serambeque, Ana Cristina Gonçalves, Ana Bela Sarmento-Ribeiro, Eunice Carrilho, Maria Filomena Botelho, Anabela Baptista Paula, Manuel Marques Ferreira

**Affiliations:** 1Institute of Endodontics, Faculty of Medicine, University of Coimbra, 3000-075 Coimbra, Portugal; ines17ferreira@gmail.com (I.F.); m.mferreira@netcabo.pt (M.M.F.); 2Institute of Biophysics, Faculty of Medicine, University of Coimbra, 3000-548 Coimbra, Portugal; mafaldalaranjo@gmail.com (M.L.); beatrizprazserambeque@gmail.com (B.S.); mfbotelho@fmed.uc.pt (M.F.B.); 3Institute for Clinical and Biomedical Research (iCBR), Area of Environment Genetics and Oncobiology (CIMAGO), Faculty of Medicine, University of Coimbra, 3000-548 Coimbra, Portugal; cmiguel.marto@uc.pt (C.M.M.); joao.casalta@gmail.com (J.C.-L.); acc.goncalves@gmail.com (A.C.G.); absarmento@fmed.uc.pt (A.B.S.-R.); eunicecarrilho@gmail.com (E.C.); 4Centre for Innovative Biomedicine and Biotechnology (CIBB), 3000-548 Coimbra, Portugal; 5Clinical Academic Center of Coimbra, CACC, 3004-561 Coimbra, Portugal; 6Institute of Experimental Pathology, Faculty of Medicine, University of Coimbra, 3000-548 Coimbra, Portugal; 7Radiation Oncology Department, Coimbra University Hospital Centre, 3000-548 Coimbra, Portugal; 8Laboratory of Oncobiology and Hematology, Faculty of Medicine, University of Coimbra, 3000-548 Coimbra, Portugal; 9Institute of Integrated Clinical Practice, Faculty of Medicine, University of Coimbra, 300-075 Coimbra, Portugal; 10Avenida Byssaya Barreto, Blocos de Celas, 3000-075 Coimbra, Portugal

**Keywords:** epoxy resins, tricalcium phosphate, cytotoxicity, cell viability, GuttaFlow^®^ bioseal, AH26^®^

## Abstract

The sealers used for root canal treatment should be biocompatible for the peri-radicular tissues, to evaluate the cytotoxic effects of GuttaFlow^®^ bioseal sealer and to compare them with AH26^®^ epoxy resin. Culture media were conditioned with the GuttaFlow^®^ bioseal and AH26^®^ pellets. MDPC-23 odontoblast cell cultures were treated with conditioned medium and serial dilutions. To evaluate the metabolic activity and cellular viability, the MTT and SRB assays were performed. To determine the production of reactive oxygen species, the DHE and DCF-DA probes were used. Cell cycle and cell-death types were assessed by cytometry, and to evaluate the mineralization capacity, the Alizarin Red S coloration was used. Statistical analysis was performed using analysis of variance (ANOVA) when normality was found and Kruskal-Wallis on the opposite case. For the comparison with normality values, the Student *t*-test was used. Cells exposed to the GuttaFlow^®^ bioseal conditioned medium maintained high metabolic activities, except at higher concentrations. Likewise, viability was maintained, but a significant decrease was observed after exposure to the highest concentration (*p* < 0.001), associated with cell death by late apoptosis and necrosis. When cell cultures were exposed to AH26^®^, metabolic activity was highly compromised, resulting in cell death. An imbalance in the production of peroxides and superoxide anion was observed. GuttaFlow^®^ bioseal showed higher biocompatibility than AH26^®^.

## 1. Introduction

Root canal treatment involves disinfection and conformation, followed by the tridimensional obturation and sealing the root canal system. This therapy aims to avoid infection and to promote the repair of the peri-radicular tissues [1]. Filling the cleaned and aseptic root canal is critical; infection could cause new lesions or prevent the repair of the existing pathology.

Root canal filling materials must have the capacity to form an airtight barrier with dentin and be biocompatible regarding peri-radicular tissues [2,3]. The use of gutta-percha cones for root canal filling is a generalized method. However, due to the insufficiency of this material to ensure adequate, hermetic, and three-dimensional canal filling, endodontic cements are used. These sealers aim to fill the irregularities between the dentin walls and the gutta cone, the lateral and accessory canals, and seal the dentinal tubules to prevent root canal infection [4,5]. Containment of the root canal sealers is important since some materials cause tissue reactions and increase inflammation [3]. For example, when extrusion of the sealer occurs, it may not be reabsorbed from the peri-radicular tissues, which may alter the healing time or cause undesirable tissue reactions [4,6,7].

The root canal sealer composition is an essential factor for its biocompatibility [6,8]. Also, biocompatibility depends fundamentally on the formation of these materials’ sealer bioproducts, but also in an indirect way of its adhesive characteristics, the stability of the cement and the area of contact between this and the soft and hard tissues [9].

The hydraulic calcium silicate cements present dimensional stability, adequate sealing capacity, and good radiopacity, bind chemically to the dental structure, and induce regenerative response [10]. They present excellent biocompatible properties due to their similarity with hydroxyapatite producing on the hydration process, components capable of inducing a regenerative response in the human body [11,12,13]. Also, these hydraulic calcium silicate cements demonstrate osteoconductive activity, promoting bone formation and an osteoinductive capacity due to their ability to bind directly to the bone to induce a bone-healing process [13,14,15]. Also, these cements present antibacterial and antifungal properties [11]. These characteristics are derived from in situ precipitation, which occurs after setting, and from having pores containing 1–3 mm diameter nanocrystals that prevent bacterial adhesion [11,13]. In addition, they are non-toxic [11] and are bio-inert materials.

GuttaFlow^®^ bioseal is a new formulation of a silicon-based sealer combined with calcium silicate particles. It contains calcium and silicate, which seem to stimulate tissue regeneration and healing potential properties [3].

AH26^®^ (Dentsply Maillefer, Baillagues, Switzerland) is an epoxy resin-based sealer, commonly used for the root canal filling with cytotoxic attributed to the release of formaldehyde as a result of a chemical setting reaction that is reduced after 24 h. Although relatively stable, endodontic cements may dissolute and, in some cases, cause or worsen tissue damage and participate in the development of periapical inflammations or maintain a pre-existing periapical lesion, delaying healing, and adversely affecting the treatment outcome [3,5,9,10]. Thus, these materials should not be toxic to hard or soft tissues, i.e., they must be biologically compatible with a positive tissue response when repairing apical lesions, particularly in the case of extravasation into the periapical area [3,5].

Cytotoxicity tests are primary biocompatibility tests that determine cell activity, inhibition of growth, cell lysis, and other effects on them caused by the tested substances [1]. In vitro cytotoxicity studies are essential to evaluate the safety of endodontic cements [6] and their chemical–biological interactions [10,16,17], to ensure the viability of peri-radicular cells and the absence of cell-death pathways such as apoptosis or necrosis [5]. To increase studies’ reproducibility, the use of standards is an added value. Regarding in vitro cytotoxicity assays, the International Standard Organization ISO 10993-5 is considered for cytotoxicity evaluation [14,18].

The present investigation aimed to evaluate the cytotoxicity of a recently introduced GuttaFlow^®^ bioseal compared to AH26^®^, in an in vitro model the MDPC-23 odontoblast cell line. Thus, the null hypothesis to be tested is that there are no differences between materials, and the alternative is GuttaFlow^®^ bioseal is less cytotoxic.

## 2. Results

### 2.1. Metabolic Activity

GuttaFlow^®^ bioseal determined a decrease of the cellular metabolic activity, with the increase of the incubation time, except for the conditioned medium 1:16. Relative to AH26^®^, a reduction of metabolic activity was also observed with increasing incubation time; furthermore, after 120 h of incubation, the cellular metabolic activity was residual (Figure 1).

Detailing the results obtained, after 24 h incubation with GuttaFlow^®^ bioseal, a significant decrease in metabolic activity to 75.03 ± 5.47% (*p* = 0.006) and 17.36 ± 5.33% (*p* < 0.001) at concentrations 1:2 and 1:1, respectively, can be observed. For a 72 h incubation period, the metabolic activity decreases to 71.04 ± 5.10% (*p* < 0.001), to 67.62 ± 7.06% (*p* = 0.018) and to 2.94 ± 1.10% (*p* < 0.001) at concentrations of 1:4, 1:2 and 1:1, respectively. At 120 h the metabolic activity decreases on the concentrations of 1:8. 1:4, 1:2 and 1:1 to 80.63 ± 4.56% (*p* < 0.001), 72.07 ± 5.93% (*p* = 0.006), 53.30 ± 2.30% (*p* < 0.001) and 1.11 ± 0.11% (*p* < 0.001), respectively.

Observing the metabolic activity of cells treated with AH26^®^, after 24 h a decrease in the metabolic activity occurred for the concentrations of 1:8, 1:4, 1:2 and 1:1, with values of 42.82 ± 7.38% (*p* < 0.001), 8.36 ± 3.07% (*p* < 0.001), 2.52 ± 0.78% (*p* < 0.001) and 2.02 ± 0.54% (*p* < 0.001), respectively. For a 72 h of incubation period, a decrease to only 12.71 ± 0.64% (*p* < 0.001), 0.93 ± 0.15% (*p* < 0.001), 0.66 ± 0.08% (*p* < 0.001) and 0.90 ± 0.12% (*p* < 0.001) was seen at the respective concentrations of 1:8, 1:4, 1:2 and 1:1. Likewise, at 120 h of incubation, the metabolic activity was less than 1% (*p* < 0.001) in all concentrations tested.

### 2.2. Cell Viability and Types of Cell Death

The viability of the GuttaFlow^®^ bioseal treated cell cultures was not affected by the treatment, except at the highest concentration tested (1:1), where a decreased to 29.82 ± 6.26% (*p* < 0.001) was observed, as shown in Figure 2A. After exposure to AH26^®^, in the concentrations 1:1 and 1:4, a decreased cell viability was seen 85.54 ± 5.50% (*p* = 0.117) and to 63.67 ± 6.66% (*p* = 0.006), respectively. Thus, both materials determined the loss of cell viability in the higher concentrations.

Figure 2B shows the types of cell death after 24 h of exposure to the biomaterials conditioned media. Both biomaterials determined a tendency for a decrease in living cells and an increase in cell death with the increase of the concentration.

GuttaFlow^®^ bioseal conditioned media at concentration 1:1 led to a loss of live cells, from 74.00 ± 4.81% to 36.33 ± 12.55% (*p* = 0.014), which was accompanied by an increase in late apoptosis/necrosis population to 7.17 ± 1.58% (*p* = 0.012). Apoptosis occurred as a result of exposure to GuttaFlow^®^ bioseal conditioned media of 1:4 (17.33 ± 4.47% to 36.5 ± 1.67, *p* = 0.036).

In the case of exposure to AH26^®^ conditioned media (1:1), living cells decrease is much more pronounced to 17.00 ± 3.44% (*p* = 0.002), and concomitantly an increase in the late apoptosis/necrosis population cells to 12.83 ± 1.78% (*p* < 0.001) occurs. Moreover, with the lower concentrations of 1:4 and 1:16, there was a decrease in living cells to 9.83 ± 3.12% (*p* < 0.001) and 52.82 ± 2.93% (*p* = 0.01). Concomitantly, cell populations in late apoptosis/necrosis increased from 1.5 ± 0.23% to 51.83 ± 16.96% (*p* < 0.001) and to 11.17 ± 3.92% (*p* = 0.008), respectively.

### 2.3. Cell Cycle

Figure 3 shows the cell cycle of the cultures conditioned by the biomaterials.

GuttaFlow^®^ bioseal at the highest concentration (1:1) determined a decrease of cells in phase G0/G1, from 57.22 ± 7.29% to 46.88 ± 3.14% (*p* < 0.001), which was accompanied by an increase in phase S population cells from 25.50 ± 9.53% to 33.66 ± 6.87% (*p* = 0.0017). Though, on exposure to AH26^®^ conditioned media at all concentrations, no significant changes were observed.

### 2.4. Oxidative Stress

GuttaFlow^®^ bioseal led to a decrease of hydrogen peroxide production to 74.66 ± 4.49% (*p* < 0.001), 21.36 ± 0.93% (*p* < 0.001) and 24.86 ± 3.75% (*p* < 0.001), for the concentrations of 1:16, 1:4 and 1:1 respectively, as observed in Figure 4.

Also, the treatment with AH26^®^ resulted in an even more pronounced decrease in the production of peroxides to 39.27 ± 1.78% (*p* < 0.001), 20.74 ± 0.68% (*p* < 0.001), and 20.17 ± 1.06% (*p* < 0.001) for the concentrations 1:16, 1:4 and 1:1, respectively.

Hydrogen peroxide production decreased in a more pronounced manner after exposure to AH26^®^ than with GuttaFlow^®^ bioseal at the concentration of 1:16 (*p* < 0.001).

Regarding the production of superoxide anion after treatment with GuttaFlow^®^ bioseal, the conditioned medium 1:1 seems to be associated with a tendency to increase the production of this byproduct, to a value of 119.49 ± 10.33% (*p* > 0.05). The conditioned media 1:4 and 1:16 determined the values of 92.96 ± 10.48% and 55.38 ± 5.29% (*p* < 0.001), respectively, the latter case corresponding to a significant decrease. Superoxide anion production by MDPC-23 cells after exposure to AH26^®^ was shown to be similar at all concentrations, i.e., at the 1:1 concentration, the superoxide production was 53 ± 3.23% (*p* < 0.001), at 1:4 concentrations, was 50.53 ± 3.10% (*p* < 0.001), and at the 1:16 concentration was 51.11 ± 2.42% (*p* < 0.001).

The intracellular concentration of superoxide anion was significantly lower after exposure to the media conditioned by the AH26^®^, at 1:4 and 1:1 concentration, than after exposure to media conditioned by bioseal GuttaFlow^®^ (*p* < 0.001).

### 2.5. Alizarin Red S

MDPC-23 cells were able to produce mineralized deposits, as observed in Figure 5A.

However, no marked differences were observed among conditions. For the highest concentration (1:1), a decrease in absorbance of 0.10 ± 0.01% to 0.06 ± 0.004% (*p* = 0.031) and to 0.06 ± 0.002% (*p* = 0.038) was observed, in the case of GuttaFlow^®^ bioseal and AH26^®^, respectively.

## 3. Discussion

After root canal preparation, the canal system needs to be filled with a suitable material for periapical wound-healing to occur. For this purpose, a new family of root canal sealers are emerging and should be evaluated for cytotoxicity [19,20]. In general, root canal sealers, when used in contact with cells and pulp tissues, exhibit variable cytotoxic effects [21].

In the literature, there is a limited number of studies that compare the cytotoxicity of GuttaFlow^®^ bioseal with the AH26^®^ sealer. The reason for selecting AH26^®^ to compare with GuttaFlow^®^ bioseal was the fact that this epoxy resin-sealer is still used in the clinic, due to their adequate properties as a sealer, radiopacity, and dimensional stability [22], despite the current epoxy gold-standard being AH Plus [23].

Well described and validated assays, such as the MTT assay, SRB, and oxidative stress evaluation, were performed to evaluate the cytotoxic effects of the materials. Relatively to metabolic activity, it was strongly influenced by the exposure time of the cell line to the epoxy resin AH26^®^. GuttaFlow^®^ bioseal maintained metabolic activities superior to 75%, except for the highest concentration of 1:1. In fact, studies by other authors indicate that GuttaFlow^®^ bioseal does not determine statistically significant differences in the number of viable cells and that the cytotoxicity of this compound depends on the incubation time [3]. Several authors also report that GuttaFlow^®^ bioseal is an endodontic sealer with higher biocompatibility compared to other cements used in current clinical practice [3,6]. Its biocompatibility may be due to the presence of bioactive components, such as calcium silicate, as well as to the absence of resin in its constitution [3,6]. However, in most of these studies, only tests to assess metabolic activity were performed, which present similar results to our research. Other more specific tests, such as the evaluation of cell viability and types of cell death, cell cycle, or assessment of the production of reactive oxygen species, have not been performed.

Regarding AH26^®^, a greater loss of metabolic activity was observed with the increase of the exposure time, and the 1:1 concentration was totally deleterious. When 120 h of exposure were reached, even low concentrations of AH26^®^ determine high cytotoxicity. Several studies showed the cytotoxic effect of AH26^®^ in different cell lines [24,25,26,27,28,29,30]. The increased cytotoxicity of this sealer has been associated with the release of formaldehyde [24,25,26,27,28,29,30], which occurs during setting due to the hexamethylenetetramine composition [27].

The SRB assay directly correlates cell viability with protein content and complements the MTT assay, which allows the identification of metabolically active cells [31,32]. The GuttaFlow^®^ bioseal, in 1:16 and 1:4 concentrations, determined a high percentage of metabolically active cells, and cell viability is in line with these results, being very similar to the control group. Consequently, the cytotoxic effect at these concentrations is considered low, as it does not exceed 30%, as recommended by ISO 10993-5 [14,18]. AH26^®^ has been shown to have significant cytotoxicity in concentrations not only 1:1 but also 1:4, with Koulaouzidou et al. having reported similar results [33].

Cell viability and types of death are dependent on the concentration of the biomaterial, i.e., the lower biomaterial concentration, the lower cytotoxicity. The evaluation of cell death to GuttaFlow^®^ bioseal at 1:16 shown to be the most similar to the control group, both in cell viability and in types of death. With an increase in concentration, there is a decrease in living cells and an increase in apoptosis first and in late apoptosis/necrosis later. Therefore, this material induces lower cell death, being less cytotoxic than AH26^®^. With the latter, a decrease in living cells to shallow values at all concentrations is observed, accompanied by a great increase in late apoptosis/necrosis.

Reactive oxygen species play an essential role in the regulation of cell signaling pathways. However, when they are in high concentrations, they can cause significant damage to the cellular structure [34,35]. Both compounds lead to a disruption of intracellular ROS concentrations 24 h after cell treatment.

Alizarin Red S staining was used to evaluate the degree of cellular mineralization, which is maintained as in control cultures. Despite the lower absorbance values obtained at the highest concentration (1:1) both materials, this is undoubtedly due to loss of cellular viability instead of impairment of the mineralization process.

The cell cycle consists of cell proliferation, in which one cell originates another with the same genetic characteristics. There were no significant differences in the AH26^®^ treated cell cultures, regarding the control. GuttaFlow^®^ bioseal in the highest concentration leads to a decrease in the G1 population and to an increase in the S-phase population, which is compatible with a cytostatic effect.

Some limitations must be considered regarding this study. First, being an in vitro study, it is a simplistic model of clinical reality. Moreover, we used dental pulp cells instead of other cells, which the sealers contact during the root canal treatment, such as periodontal ligament cells, osteoblasts, or apical papilla stem cells. However, MDPC-23 cells are considered a good model for evaluating cytotoxicity [1]. Another limitation was that the endodontic sealers were tested after setting and were not compared with a freshly mixed material. Despite this limitation, because freshly mixed materials are generally more cytotoxic, we tested high concentrations and serial dilutions to simulate tissue diffusion. Also, this work provides a comprehensive analysis with several complementary methods. To confirm the beneficial effects of GuttaFlow^®^ bioseal, used as root canal sealer, further studies are needed, particularly considering direct and indirect contact.

## 4. Materials and Methods

### 4.1. Cell Culture

The MDPC-23 is a spontaneously immortalized odontoblast-like cell line, which was graciously offered by the University of Michigan. The cells were cultured in Dulbecco’s Modified Eagle’s Medium (DMEM) (Sigma D-5648, Sigma, Kawasaki, Japan) supplemented with 1% sodium pyruvate (Gibco 11360), 1% antibiotic (100 U/mL penicillin and 10 µg/mL streptomycin; Sigma A5955) and 5% fetal bovine serum (FBS) (Sigma F7524), and maintained at 37 °C in a humidified atmosphere with 95% air and 5% carbon dioxide. To prepare cell suspensions, the cell cultures were detached by washing with phosphate-buffered saline (PBS) and incubating with trypsin-EDTA (Sigma-Aldrich, Saint Louis, MO, USA).

### 4.2. Pellets and Extracts Preparation

GuttaFlow^®^ bioseal (Coltené, Langenau, Germany) was tested in comparison with the standard root canal sealer AH26^®^ (Dentsply, Konstanz, Germany). The materials composition and setting times are described in Table 1. The sealers were prepared according to the manufacturer’s instructions and shaped in a PVC mold with 1.5-mm-thickness and a diameter of 3 mm. After the setting times, the pellets obtained were removed from the mold and placed in 6-well plates to be sterilized under UV light for 20 min on each face.

In the laminar flow hood, the pellets were added to a falcon tube with DMEM and placed in the incubator for 24 h. For this, a surface area per volume of medium of 250 cm^2^/mL was considered. After the incubation time, the initial solution obtained was considered to correspond to the 1:1 concentration, and from this, four successive dilutions (1:2, 1:4, 1:8, 1:16) were prepared [36].

### 4.3. MTT Assay

The MTT (3-(4,5-dimethylthiazol-2-yl)-2,5-diphenyltetrazolium bromide) (Sigma M2128) assay was performed after 24, 72, and 120 h of incubation of the cell cultures with the conditioned media. For this, the cell culture medium was removed, the plate was washed with PBS, and the MTT solution was placed in each well, in a concentration of 0.5 mg/mL. The plate was incubated overnight protected from light [3,6,10,34,37,38].

The next day acidic isopropanol (0.04 M HCl) was added [37,38,39], and each plate was stirred for 20 min to ensure complete solubilization [3,10,39]. The absorbance was read on the EnSpire^®^ spectrophotometer considering a wavelength of 570 nm and a reference value of 620 nm [3,6,38,39].

### 4.4. SRB Assay

The sulforhodamine B (SRB) assay was performed after 24 h of incubation of the cell cultures with the conditioned media. For this, the medium was removed, the cells were washed with PBS and fixed with a solution of 1% acetic acid in methanol for 1 h at 4 °C. Subsequently, the solution of 0.5% SRB in 1% acetic acid was added to dye the cells, and the plates were incubated for 30 min, at room temperature, in the dark. Then, plates were washed, dried, and a solution of 10 mM Tri-NaOH, pH = 10, was added and homogenized. To read the absorbance at a wavelength of 540 nm, the EnSpire^®^ spectrophotometer was used.

### 4.5. Types of Cell Death

After incubating the cell cultures for 24 h with the conditioned media, the cells were detached, centrifuged, and washed with PBS. Cells were labeled in binding buffer [consisting of 0.01 M Hepes (Sigma, H7523), 0.14 mM NaCl (Sigma, S7653) and 0.25 mM CaCl_2_ (Sigma, C4901)], with annexin V (labeled with fluorescence isothiocyanate) and propidium iodide as recommended by the manufacturer of the kit (Immunostep ANXVFKIT Immunotech and KIT Immunotech). After vortex and incubation for 15 min, at room temperature, in the dark, cells were analyzed by flow cytometry in the FACSCalibur cytometer (BD Biosciences, Qume Drive, San Jose, CA, USA).

### 4.6. Cell Cycle

After 24 h of incubation with the conditioned media, cells were detached, centrifuged (2500 rpm for 5 min), and washed with PBS. The pellet was fixed in a vortex with 70% ethanol and incubated for 30 min at 4 °C. The suspension was washed with PBS and labeled with propidium iodide for 15 min, at 37 °C, in the dark. Cells were analyzed in the FACS Calibur cytometer.

### 4.7. Oxidative Stress

The probes dihydroethidium (DHE) and 2′,7′-dichlorodihydrofluorescein diacetate (DCFH_2_-DA) were used. The cells were detached, centrifuged, and incubated with the probes in PBS. DHE was used in a concentration of 5 µM for an incubation time of 15 min. DCF-HA probe was added to make up a concentration of 5 µM and incubated for 45 min at 37 °C. After the incubation time, the suspension was washed by centrifugation, suspended in PBS, and analyzed in the fluorimeter using the excitation and absorbance wavelengths of 530 nm and 645 nm for the DHE probe and 485 nm and 528 nm for the DCFH2-DA.

### 4.8. Mineralized Nodules Quantification

The cell cultures were washed three times with PBS and fixed with 4% paraformaldehyde for 15 min at room temperature. Three more washes were performed. The cell cultures were stained with a 40 mM alizarin red staining solution (pH 4.2, A5533 Sigma-Aldrich) for 20 min at 37 °C [36]. Washings were performed until complete removal of the dye. After the acquisition of photomicrographs, an extraction solution, composed of 10% acetic acid and 20% methanol, was used for 40 min under stirring at room temperature, followed by the absorbance reading at 490 nm using the EnSpire^®^ spectrophotometer.

### 4.9. Statistical Analyzes

The results are presented as mean ± standard error of at least three experiments performed in duplicate. Statistical analysis was performed using IBM^®^ SPSS^®^ software version 20. The Shapiro-Wilk test was used to evaluate the normal distribution of the quantitative variables, given the sample size. The results of the treated cell cultures were compared with the control cell cultures using the one-way analysis of variance (ANOVA) in cases where there was a normal distribution and homogeneity of variances. When the contrary was found, the Kruskal-Wallis test was performed. Multiple post-hoc comparisons were then performed between experimental groups using appropriate parametric or non-parametric tests, with significance adjustment according to the Bonferroni correction. Comparisons with the normalization value of 100% were performed using the Student’s *t*-test for one sample. A significance value of 0.05 was considered for all comparisons.

## 5. Conclusions

In conclusion, GuttaFlow^®^ bioseal shows a lower cytotoxic profile compared to the AH26^®^ sealer. The AH26^®^ leads to loss of cellular viability, associated with the activation of types of cell death, in particular, late apoptosis and necrosis. The cytotoxicity of these materials is dependent on the time of exposure and concentration of the biomaterial, whose cytotoxic effect may be related to the production of ROS.

## Figures and Tables

**Figure 1 molecules-25-04297-f001:**
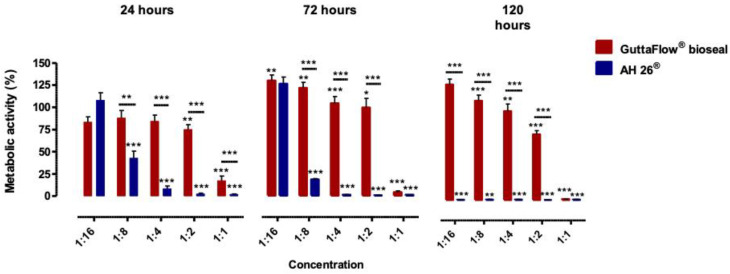
Metabolic activity of the MDPC-23 cell cultures after 24, 72, and 120 h of incubation with the biomaterials conditioned media. The results are presented as the mean and standard error of a minimum of six independent assays. Statistically significant differences to the control or between conditions are represented by a *, where * means *p* < 0.05, ** means *p* < 0.01, and *** means *p* < 0.001.

**Figure 2 molecules-25-04297-f002:**
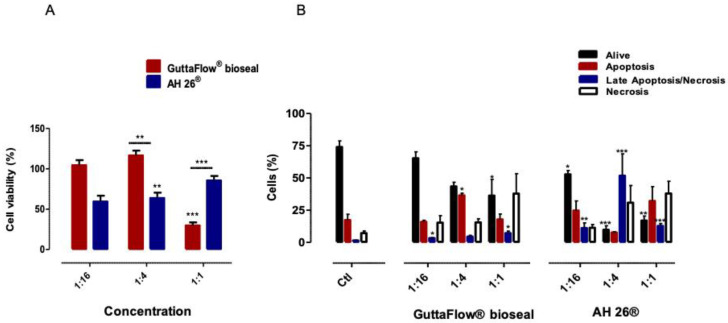
Cell viability (**A**) and types of cell death (**B**) of the MDPC-23 cell cultures after 24 h of incubation with the biomaterials conditioned media. The results are presented as the mean and standard error, with a minimum of six independent assays. Statistically significant differences to the control or between conditions are represented by a *, where * means *p* < 0.05, ** means *p* < 0.01, and *** means *p* < 0.001.

**Figure 3 molecules-25-04297-f003:**
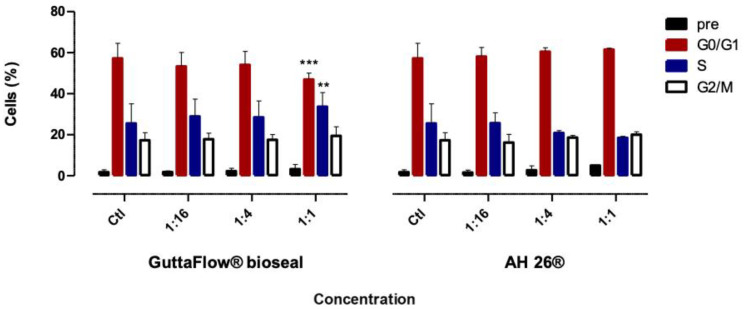
Cell cycle phases of the MDPC-23 cell cultures after 24 h of incubation with the biomaterials conditioned media. The results are presented as the mean and standard error, with a minimum of six independent assays. Statistically significant differences to the control or between conditions are represented by a *, where ** means *p* < 0.01, and *** means *p* < 0.001.

**Figure 4 molecules-25-04297-f004:**
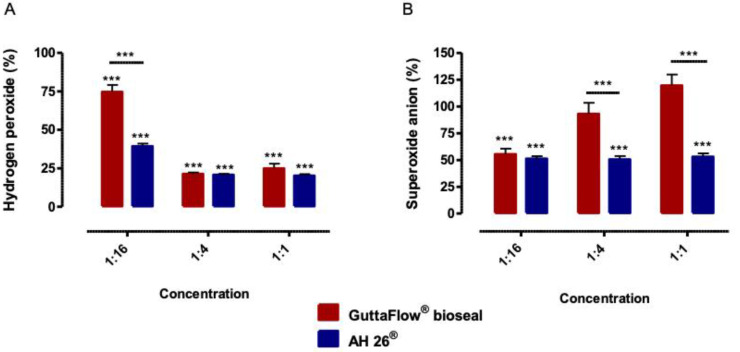
Intracellular production of hydrogen peroxide (**A**) and superoxide anion (**B**) of the MDPC-23 cell cultures, after 24 h of incubation with the biomaterials conditioned media. The results are presented as the mean and standard error, with a minimum of six independent assays. Statistically significant differences to the control or between conditions are represented by a *, where *** means *p* < 0.001.

**Figure 5 molecules-25-04297-f005:**
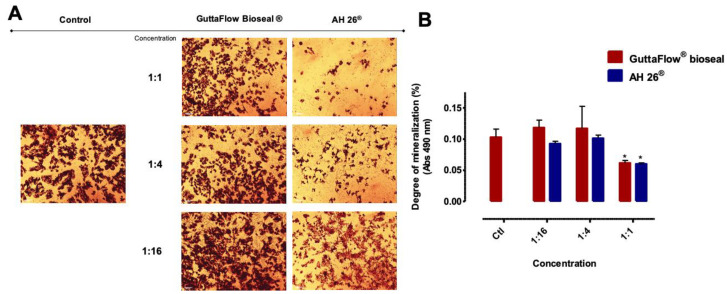
MDPC-23 stained cells with Alizarin Red S after 24 h of incubation with the biomaterials conditioned media (**A**) and staining quantification (**B**). The control group represents cells in culture in Dulbecco’s Modified Eagle’s Medium (DMEM) with 5% fetal bovine serum (FBS). The results are presented as the mean and standard error, with a minimum of six independent assays. Statistically significant differences to the control or between conditions are represented by a *, where * means *p* < 0.05.

**Table 1 molecules-25-04297-t001:** Materials composition, setting times, lot, and manufacturer.

Materials	Composition	Setting Time	Lot	Manufacturer
GuttaFlow^®^ Bioseal	Gutta-percha power, polydimethylsiloxane, platinum catalytic agent, zirconium dioxide, bioactive ceramic glass	9–15 h	H71011; I33299; I46491; I14741	Coltené, Langenau, Germany
AH26^®^	Bismuth oxide, methanamine, silver, titanium dioxide, epoxy resin	25–30 min	1703000255; 1701000094	Dentsply, Konstanz, Germany

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
