# Peer review of "GuttaFlow® Bioseal Cytotoxicity Assessment: In Vitro Study"

_molecules, 2020, doi:10.3390/molecules25184297_

Round 1

Reviewer 1 Report

none

Reviewer 2 Report

The paper has been improved and is acceptable now

This manuscript is a resubmission of an earlier submission. The following is a list of the peer review reports and author responses from that submission.

Round 1

Reviewer 1 Report

Unfortunately, the value of the investigation is considerably limited due to some severe shortcomings. The concerns in detail:

  • Abstract: statistical test used should be given. Sample size is not mentioned. P-values must be given
  • Throughout the entire manuscript it is wrongly stated that both sealers are cements. However, GuttaFlow bioseal is a silicone-based sealer and AH 26 is an epoxy resin.
  • Introduction: this section should be re-written by someone more familiar with the endodontic terminology: “endodontic treatment” should read “root canal treatment” as this is a more precise description; “airtight” must be “fluid tight”; “powder with hermetic properties” is a wrong and misleading formulation.
  • Composition of AH 26 must be given.
  • A null-hypothesis should be formulated.
  • The main shortcoming of this study is the use of AH 26 instead of the current gold-standard AH Plus. AH 26 is no longer the epoxy-resin sealer of first choice. It is well known that that the biocompatibility of this sealer is less than ideal. This convey an impression that a kind of bias exists: comparing a relatively new sealer with a well-studied sealer that exerts some cytotoxic effects does not provide relevant results.
  • Sample size calculation is missing.
  • Provide setting times for both sealers.
  • Fully set sealers were used although it is well known that nearly all sealers display a more pronounced cytotoxicity during the setting process compared to their fully set state.

Reviewer 2 Report

RE: GuttaFlow®bioseal Cytotoxicity Assessment: In Vitro 2 Study.

This study aims to investigate the Cytotoxicity of GuttaFlow®bioseal.

The paper is not original and has not interest because many results are known.

The materials and methods are very poor.

Line 76 demonstrate osteoconductive activity…….

I suggest expanding this section after reading the following article                             

PMID: 2960302, PMID: 28233601 and

What is the hypothesis of the study?

An in vitro study is a simple way to test some hypothesis. This methodology could provide valuable information for the clinician, but it represents a simplification of the clinical reality. The authors must disclose all the possible limitations of the present model and discuss them. The limitations of this study and recommendations for future research generalized from the observations of this study need to be listed and discussed in more detail (Discussing study limitations in reports of biomedical studies – the need for more transparency, Puhan et al, 2012): Report on all limitations that may have affected the quality of the evidence being presented, including aspects of study design and implementation. Readers depend on a candid communication by the authors and may get the impression that the investigators were naive if they are not reported.

Please add more information on the limits of this research.